# The Bor1 elevator transport cycle is subject to autoinhibition and activation

Yan Jiang [1,2] ✉ & Jiansen Jiang [1] ✉

Boron, essential for plant growth, necessitates precise regulation due to its potential toxicity. This regulation is achieved by borate transporters (BORs), which are homologous to the SLC4 family. The *Arabidopsis thaliana* Bor1 (AtBor1) transporter from clade I undergoes slow regulation through degradation and translational suppression, but its potential for fast regulation via direct activity modulation was unclear. Here, we combine cryo-electron microscopy, mutagenesis, and functional characterization to study AtBor1, revealing high-resolution structures of the dimer in one inactive and three active states. Our findings show that AtBor1 is regulated by two distinct mechanisms: an autoinhibitory domain at the carboxyl terminus obstructs the substrate pathway via conserved salt bridges, and phosphorylation of Thr410 allows interaction with a positively charged pocket at the cytosolic face, essential for borate transport. These results elucidate the molecular basis of AtBor1's activity regulation and highlight its role in fast boron level regulation in plants.

Boron is an essential micronutrient for plant growth, participating in various biological processes such as cell wall synthesis and structure, membrane integrity and function, stimulation of reproductive tissues, biosynthesis and transportation of plant hormones, ion transport through the membrane, cell division and elongation, protein cytoskeletal function and metabolism of nucleic acids, carbohydrates, proteins, and antioxidants (ascorbic acid and polyphenols)[1–6]. Insufficient boron levels can significantly impact plant physiology, particularly root growth, pollen germination, and fruit formation, ultimately affecting crop yield and quality[7,8]. Studies have demonstrated that inadequate boron levels negatively impact the productivity of 132 crops in over 80 countries, especially in regions characterized by heavy rainfall and sandy or alkaline soils[8,9]. Plant roots absorb boron from the soil in the form of boric acid–a small and uncharged molecule that can readily permeate biological membranes. Passive diffusion of boric acid is sufficient to meet plant boron requirements when available in ample quantities[10]. However, when boron availability is limited, plants use boric acid channels belonging to the major intrinsic protein family and BORs, which are homologous to the SLC4 family to absorb boron from the soil[2,10]. Conversely, because boron is toxic at high

concentrations[10], causing changes to cellular metabolism, DNA damage, and tissue necrosis[11,12], BORs can also extrude excess boron from plant tissue and impart tolerance to high boron conditions[3,13].

The *A. thaliana* genome contains seven BOR-encoding genes (AtBor1–7), which are classified as either clade I (AtBOR1–3) or clade II (AtBOR4–7) according to amino acid sequence similarity[14]. AtBor1 is abundantly expressed in root tip cells when the supply of boron is limited. Its polar membrane localization at the stele side of root cells facilitates directional transport of boron toward the xylem[15]. However, when boron is in ample supply, AtBor1 is ubiquitinated and transported to the vacuole for degradation via the endocytic pathway[16,17]. In addition, higher boron supply triggers further downregulation through translational suppression[18], preventing the accumulation of toxic boron levels in plants. A second line of defense against high boron levels is provided by the clade II transporter AtBor4, which preferentially localizes to the soil side of root cells and is upregulated during high boron conditions[3,14,19]. These regulatory mechanisms provide relatively slow adaptations to both low and high boron levels. In humans, some SLC4 transporters, such as sodium-dependent bicarbonate transporters NBCe1 and NDCBE, are under activity regulation by its N- or C-terminal regions[20–27]. However,

[1]Laboratory of Membrane Proteins and Structural Biology, Biochemistry and Biophysics Center, National Heart, Lung, and Blood Institute, National Institutes of Health, Bethesda, MD, USA. [2]Present address: Transporter Biology Group, School of Medical Sciences, Faculty of Medicine and Health, University of Sydney, Sydney, New South Wales, Australia. ✉e-mail: yan.jiang@sydney.edu.au; jiansen.jiang@nih.gov

it remains unclear whether plant BORs are subject to direct and fast regulation of their activity.

In this study, we combine cryo-EM, mutagenesis, and yeast complementation assay to characterize the structure and activity regulation of full-length AtBor1. Our 2.15 Å structure of the wild-type transporter is solved in the inward-autoinhibited (IAI) conformation and reveals previously unknown detail, including C-terminal (Ct) domain swapping within the AtBor1 dimer. The Ct domain interacts with the transmembrane domain through conserved salt bridges and obstructs substrate entry from the cytosolic side, rendering the transporter domains in an autoinhibited state. We find this autoinhibition exists in clade I BORs, but not clade II BORs. Interestingly, we also capture an AtBor1 mutant (R637E/E641R/R643E) in not only the active inward-facing (IF) conformation, but also the occluded conformation and the hybrid IF/occluded conformation, the latter comprising one protomer in an IF conformation and the other in an occluded conformation. By uncovering the structure of the intracellular domain located between transmembrane (TM) helices 10 and 11, we observe that phosphorylation of Thr410 within the intracellular domain allows this residue to interact with a pocket of positive charge at the cytosolic face of the transmembrane domain. Furthermore, we reveal that this phosphorylation is essential for borate transport. Together, our structural and functional characterization of AtBor1 reveals a direct regulatory mechanism in plant BOR transporters and its underlying molecular basis and offers insights into the elevator mechanism of substrate transport in SLC4 transporters.

## Results

### Full-length AtBor1 is a homodimer with Ct domain swapping

To identify potential regulatory domains in AtBor1, we sought to determine high-resolution structures of the full-length protein to visualize intracellular regions that are likely involved in regulation but have eluded structural determination. Full-length AtBor1 was over-expressed in *Saccharomyces cerevisiae*, purified in phosphatidylcholine lipid nanodiscs (Supplementary Fig. 1), and the structure determined at 2.3 Å using cryo-EM (Supplementary Fig. 2). Particle expansion, 3D classification, and local refinement of one protomer improved the resolution to 2.15 Å (Supplementary Fig. 2), allowing the unambiguous building of an atomic model with robust side chain densities, doughnut-shaped aromatic rings, and water molecules in the cryo-EM density map (Supplementary Figs. 3, 4).

AtBor1 was solved as a homodimer, resembling prior Bor1 structures[28,29] as well as those of the structurally-related proteins AE1[30–33], AE2[34], NBCe1[35], UraA[36], SLC26A9[37–39], NDCBE[40], and BTR1[41] (Fig. 1a, b). The TM domain of one AtBor1 monomer comprises 14 TM helices that form two distinct subdomains–the Gate and the Core. The core is formed from TM helices 1–4 and 8–11 and contains the putative substrate-binding site. The gate is formed from TM helices 5–7 and 12–14 and provides the entire dimerization interface (Fig. 1). Several patches of non-protein densities are located in the gap between the two monomers. Phospholipid molecules can be fitted in those densities, indicating that protein-lipid and lipid-lipid interactions are also involved in dimerization (Fig. 1a). Importantly, we were able to resolve the Ct domain and construct the model up to residue 645. This revealed that the Ct domain comprises two α helices (H11 and H12) connected by a loop, followed by a rigid hairpin that deeply inserts into the cytosolic cavity of the neighboring protomer in a domain-swapped manner to block the permeation pathway like a "plug" (Fig. 1b, Figs. 2a, b and Supplementary Movie 1). Therefore, we define our full-length AtBor1-structure as the IAI conformation. This structure provides an opportunity to explore the role of the Ct domain in regulating AtBor1 activity.

### The Ct domain autoinhibits the transport function

To investigate the role of the Ct domain, we examined the interactions that the hairpin forms, particularly those with the cytosolic cavity in the neighboring protomer (Fig. 2a, b). The hairpin interacts with the TM5, the H5 (peripheral helix 5), the TM12, the $L_{12\text{-}H7}$ (the loop that connects TM12 and H7), and the $L_{H10\text{-}H11}$ of its neighbor protomer. Specifically, Arg637, Glu641, Arg643 and His644 on the Ct hairpin form salt bridges with Glu502 on TM12, Arg153 on TM5, Glu604 on $L_{H10\text{-}H11}$ and Glu323 on H5, respectively (Fig. 2c, e). Besides, Ser638 and Thr645 on the Ct hairpin forms polar interactions with Gly506 on $L_{12\text{-}H7}$ and Glu604 on $L_{H10\text{-}H11}$, respectively (Fig. 2c, e). Interestingly, Arg639 on the Ct hairpin does not form direct polar interactions with surrounding residues, but its guanidinium group forms a polar interaction with a nearby water molecule, which, in turn, forms a polar interaction with Arg153 on TM5 (Fig. 2d). In addition, Asp632 on H12, situated just prior to the Ct hairpin, established salt bridges with Arg222 on TM7 and Lys366 on TM10 (Fig. 2e).

These various interactions anchor the Ct hairpin to the cytosolic cavity between the gate and the core in the neighboring protomer. This hinders the tetrahydroxy borate anion $[B(OH)_4]^-$ from reaching its putative binding site on TM3 and TM10[28], suggesting that H12 and the Ct hairpin function as an autoinhibitory domain (AID) (Fig. 2f) to inhibit the transport activity of AtBor1.

We tested this hypothesis using a genetic complementation assay that measured the growth of *S. cerevisiae* on boric acid-supplemented plates upon expression of a series of AtBor1 expression plasmids with different Ct domain truncations. Growth of a *S. cerevisiae* Bor1 (ScBor1)-deficient strain (YO1169) on 80 mM boric acid was restored by transformation with wild-type ScBor1 (Fig. 2g), but not with wild-type AtBor1, consistent with previous studies[3,28]. However, an AtBor1 Ct domain truncation from the H12 (AtBor1Δ627), was able to complement and rescue yeast growth (Supplementary Fig. 5a). Sequence alignment revealed high conservation in the Ct hairpin region among plant BOR transporters (*A. thaliana*, rice, and wheat), in particular, a "TRSRGE" motif ($T_{636}RSRGEFRH_{644}$; Fig. 2f, Supplementary Fig. 6). Indeed, by narrowing the Ct truncations, we found that TRSRGE, and even shorter SRGE, truncations facilitated complementation and rescue of yeast growth (Supplementary Fig. 5a). This was not the case for truncations after the Ct hairpin (AtBor1Δ651; Supplementary Fig. 5a). These assays support our hypothesis that the Ct hairpin autoinhibits transport activity in AtBor1.

To identify the key residues involved in autoinhibition, we mutated residues that use polar interactions to anchor the Ct hairpin to Ala. Arg637A, Arg643A and His644A successfully complemented and rescued yeast growth in 80 mM boric acid, whereas Ser638A and Glu641A did not (Fig. 2g, Supplementary Fig. 5a). Unexpectedly, Glu641A completely abolished yeast growth at lower boric acid concentrations (40 and 60 mM), in contrast to wild-type AtBor1 (Supplementary Fig. 5a). However, mutation of this acidic Glu residue to the neutral residue Gln rescued yeast growth up to 80 mM boric acid (Fig. 2g, Supplementary Fig. 5a). Because previous work suggested that Asp311 on TM8 is a putative proton-binding site[42], and Glu641 is the only negatively charged residue in the Ct hairpin, we wondered whether Glu641 might be similarly involved in proton coupling and Glu641Q thus mimics a constitutively protonated state. We therefore mutated charged residues on the Ct hairpin to oppositely charged residues (Arg637E, Glu641R and Arg643E) and observed complementation and rescue of yeast growth in 80 mM boric acid (Supplementary Fig. 5a). This indicates that autoinhibition is mainly dependent on electrostatic interactions in the Ct hairpin. Another charged residue on the Ct hairpin, Arg639, is highly conserved in plant BOR transporters (Fig. 2f, Supplementary Fig. 6). Mutation of this residue to Ala, Glu and Lys complemented and rescued yeast growth in 80 mM boric acid (Fig. 2g, Supplementary Fig. 5a), indicating that the tripartite Arg639–water molecule–Arg153 interaction plays an important role in anchoring the Ct hairpin between the gate and the core (Fig. 2d). In the middle of the hairpin, Thr636 forms extensive polar interactions with the main chains of Gly640 and Glu641 to stabilize the hairpin (Fig. 2c). We also

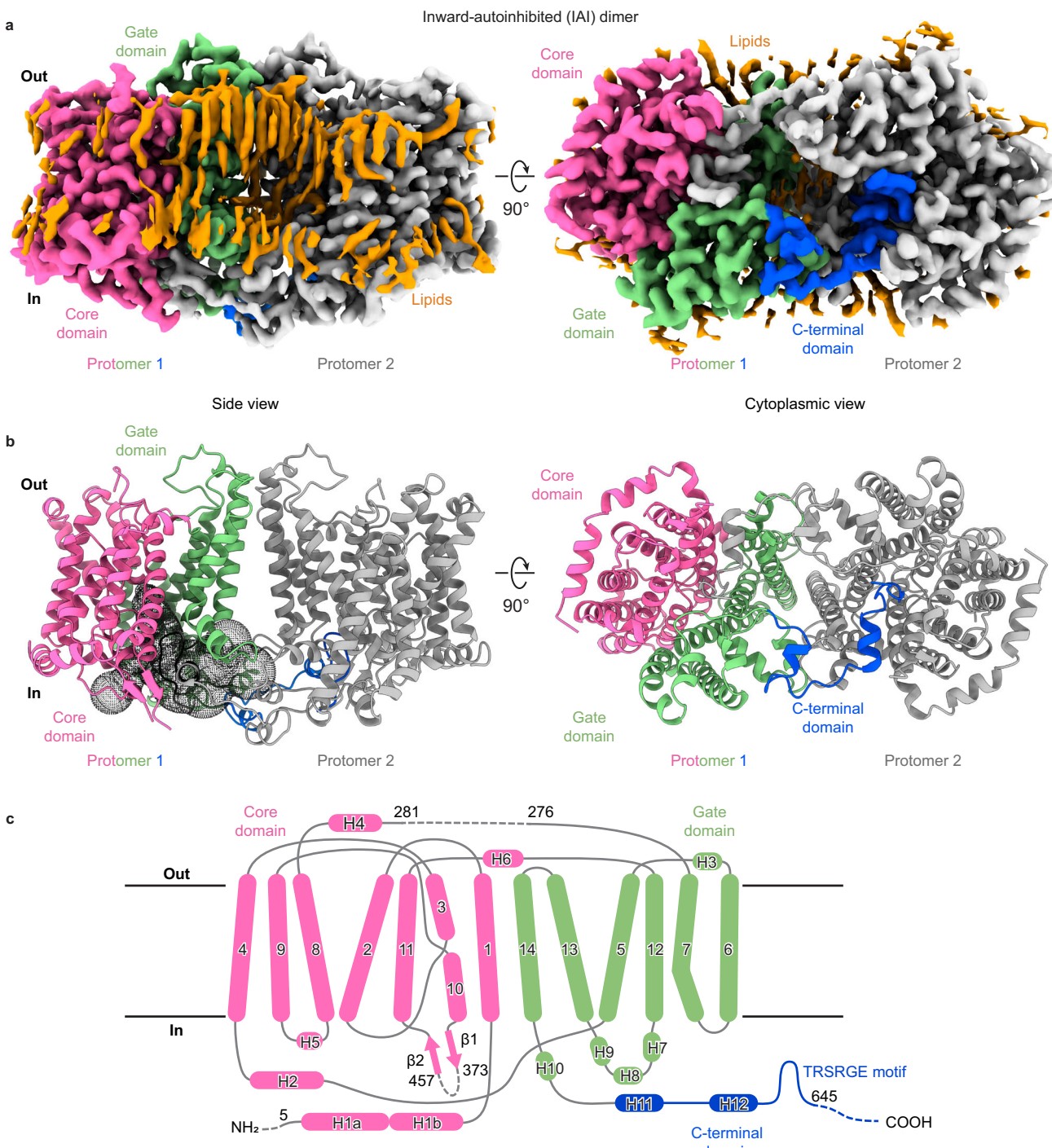

**Fig. 1 | Structure of the wild-type AtBor1 dimer in the inward-autoinhibited conformation. a** 2.3 Å cryo-EM density map of wild-type AtBor1 dimer showing side and cytoplasmic views. One protomer is colored gray and the other is colored by domains (core, pink; gate, green; Ct, blue). Visible densities of lipid molecules are colored gold. **b** Atomic model of the wild-type AtBor1 dimer in side and cytoplasmic views. Domains are colored as in (**a**). The permeation pathway in the absence of Ct occlusion is illustrated as dotted surfaces on one of the protomers. **c** Topology of the secondary structures in a wild-type AtBor1 protomer. Transmembrane helices are labeled as 1–14, and peripheral helices and β strands are labeled as H1–12 and β1–2, respectively. Secondary structures are colored by domains. Dotted lines indicate unresolved regions.

observed complementation and yeast growth for mutations of Thr636 and Gly640 to Ala (Fig. 2g, Supplementary Fig. 5a), indicating the importance of their role in maintaining the shape of the Ct hairpin (Fig. 2c).

Previous work surmised that the failure of full-length AtBor1 to complement and rescue yeast growth in high (80 mM) boric acid concentrations was because AtBor1 only exports borate at lower concentrations, which are too low to cause toxicity to *S. cerevisiae*[28]. AtBor4, on the other hand, successfully complemented and rescued yeast growth in 80 mM boric acid[3] because it extrudes boron from tissues in boron-toxic conditions. In addition, a previous study has shown that clade I BOR transporters, including AtBor1, cannot rescue yeast growth in high boric acid conditions, whereas clade II transporters, except for AtBor6 and OsBor4, can rescue yeast growth under

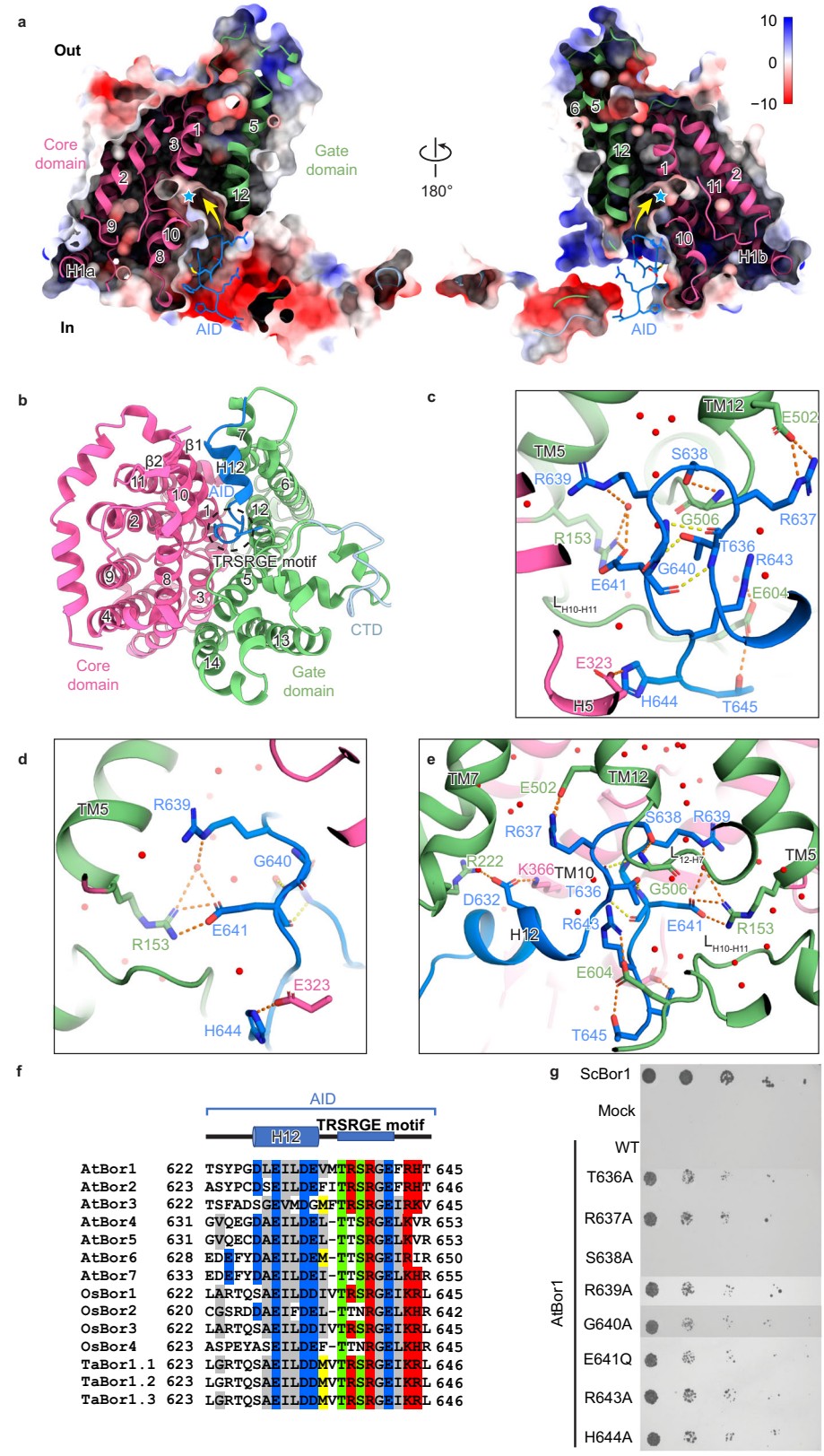

these conditions[43]. Sequence alignment shows high conservation of the Ct hairpin region, in particular the TRSRGE motif, between the AtBor1-3 from clade I, while the Ct hairpin region of the AtBor4-7 from clade II is short of one amino acid that is positioned between H12 and the TRSRGE motif and the first Arg is replaced to Thr in the TRSRGE motif (Fig. 2f). The differences on these two residues show a consistent pattern between clade I and clade II transporters, suggesting they may

play a critical role in differentiating the activity regulation mechanisms between two clades. Other clade I transporters, including OsBor1 and TaBor1.1[14], also have the same TRSRGE motif (Fig. 2f). Interestingly, we found that replacing the TRSRGE motif and its preceding methionine residue (MTRSRGEFRH sequence) in AtBor1 with the corresponding region in AtBor4 (TTSRGELKVR) complemented and rescued yeast growth in 80 mM boric acid (Supplementary Fig. 5a). These results

**Fig. 2 | Interactions between the autoinhibition domain and transmembrane domains. a** Cross-section of the wild-type AtBor1 protomer. Coulombic electrostatic surfaces are shown with a color scale from −10 to 10 kcal/(mol·e). Ribbon views of the core and gate domains are colored in pink and green, respectively. The AID (colored in blue) is from the other protomer in a domain-swapped manner. Cyan stars mark the substrate binding site between TM3 and TM10. Yellow double-headed arrows indicate the substrate pathway blocked by the AID. **b** Cytoplasmic view of a wild-type AtBor1 protomer showing the TRSRGE motif of the AID (blue), which forms a loop that inserts into the substrate entrance between the core domain (pink) and the gate domain (green). **c**–**e** Close-up views of the interactions between the AID (blue) and transmembrane domains (core domain, pink; gate domain, green). Hydrogen-bonds between the AID and transmembrane domains, and between the AID and water molecules, are shown as orange dashed lines. Hydrogen-bonds within the AID are shown as yellow dashed lines. **f** Sequence alignment of the AID of different plant Bor transporters (At, *Arabidopsis thaliana*; Os, *Oryza sativa* (rice); Ta, *Triticum aestivum* (wheat)). **g** Yeast complementation assay in 80 mM boric acid shows that a single-residue mutation of the TRSRGE motif is sufficient to abolish autoinhibition.

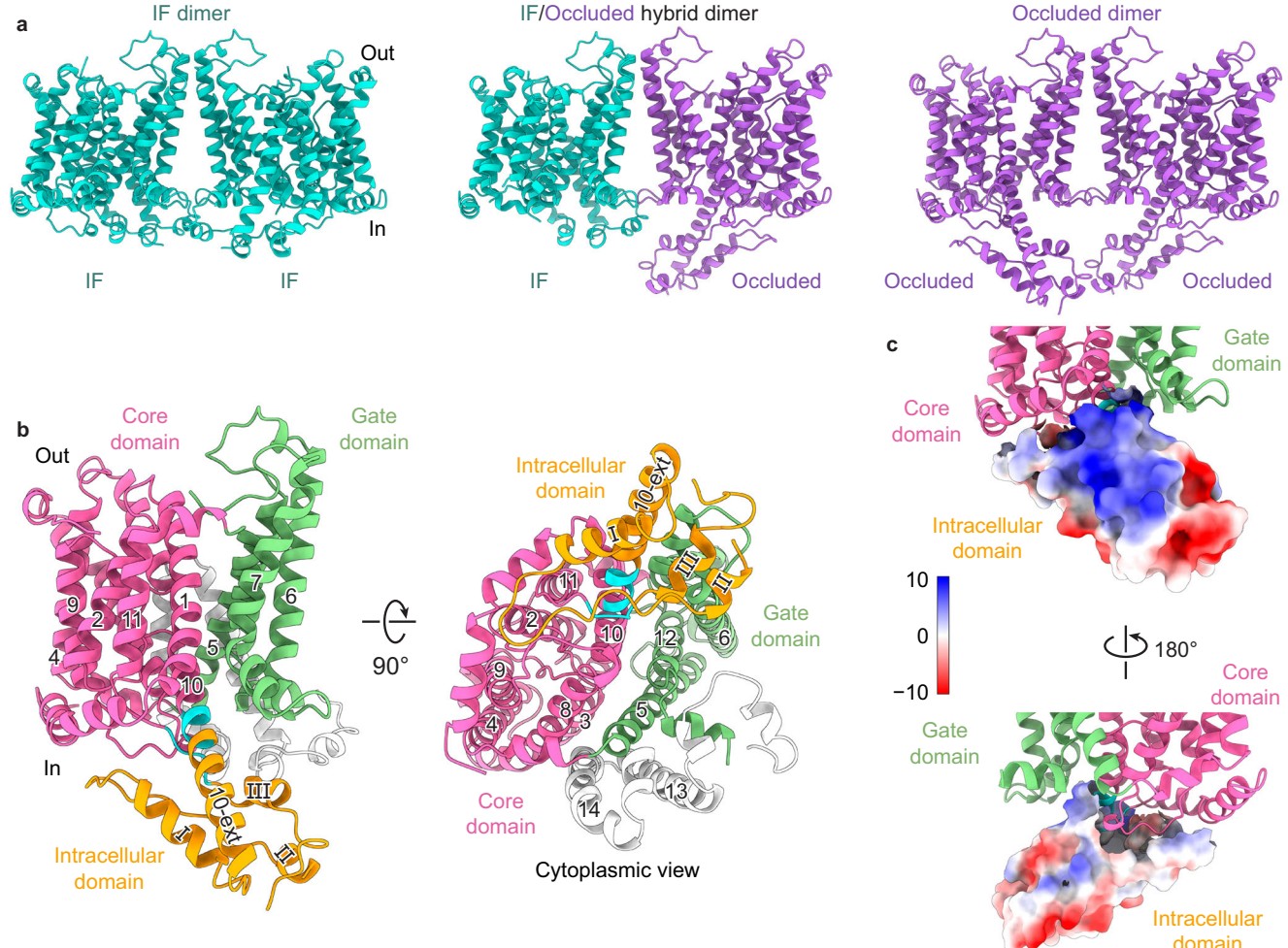

**Fig. 3 | Cryo-EM structures of AtBor1 in different active states. a** Three structures of the AtBor1$_{active}$ dimer solved from one sample using cryo-EM: IF dimer (left), IF/occluded heterodimer (middle), occluded dimer (right). **b** Atomic model of an occluded AtBor1$_{active}$ protomer. The core and gate domains are colored in pink and green, respectively. The intracellular domain in the occluded conformation is colored in gold. Residues 369–373 and 457–460 (colored in cyan) form short β-sheets (β1 and β2) between TM10 and TM11 in the IAI conformation and switch to an α-helix or loop, respectively, in the occluded conformation. TM13, TM14, and the Ct region (colored in gray) are not resolved in the occluded conformation. **c** Coulombic electrostatic surfaces of the intracellular domain in the occluded conformation. Color scale, kcal/(mol·e).

establish a Ct autoinhibitory mechanism in clade I transporters that allows them to dynamically adjust transport activity in response to varying boron conditions, thus safeguarding shoots from both boron deficiency and toxicity.

## AtBor1 protomers function independently within the dimer

We wondered whether additional regulatory mechanisms might be involved in AtBor1 function and therefore sought to capture different conformations beyond the IAI conformation of full-length, wild-type AtBor1. By making 26 mutations and using a yeast complementation assay to determine transport activity, we identified active mutations for structural studies. The protein yields of these mutant AtBor1s were

more than 10 times lower than wild type, but we obtained a stable protein sample from an AtBor1 active mutant (R637E/E641R/R643E) using LMNG detergent (Supplementary Fig. 7). We defined this mutant as AtBor1$_{active}$ and identified three distinct active conformations from 3D reconstructions following cryo-EM (Supplementary Fig. 8).

The first class was refined to 3.02 Å and resembled the IAI structure (Supplementary Fig. 9a). All 14 TM helices were well resolved in the structure, but the Ct domain was disordered (Fig. 3a, left). In the absence of the AID, the putative substrate binding site on TM3 and TM10 is accessible from the cytosolic site, categorizing this class as an IF conformation (Fig. 3a, left). The second class revealed a unique structure that to our knowledge has not been previously reported

(Supplementary Fig. 9c). The densities for TM13 and TM14 are disordered, so the atomic model was built to residue 516 from a dimer structure at 2.77 Å resolution and a protomer structure at 2.55 Å resolution with robust side chain densities (Fig. 3a, right, Supplementary Fig. 9d, e and Supplementary Fig. 10). The putative substrate binding site cannot be accessed from either the cytosolic or the extracellular sites, defining this class as an occluded conformation. The third class was solved at a resolution of 2.98 Å (Supplementary Fig. 9b) and represents a hybrid dimer with one protomer in an IF conformation and the other in an occluded conformation (Fig. 3a, middle), indicating that the two monomers can work independently of each other. This corresponds well with prior functional data from the SLC4 transporter NBCe1-a showing that each protomer has independent transport activity[44]. Our hybrid IF/occluded structure thus suggests a transport mechanism involving two functionally independent protomers.

### Phosphorylation of Thr410 of the intracellular domain is required for borate transport

In the occluded structure of AtBor1, we were able to visualize the 100-residue insertion between TMs 10 and 11 that is present in AtBor1, but not other mammalian SLC4 transporters, and which is disordered in the previous AtBor1 structure[28] and our IF and IAI structures. This region, which we define as the intracellular domain, is well refined in our occluded structure (Fig. 3b), forming three helices that we name helix I, II and III. Interestingly, residues 369–373 and 457–460, which form short β-sheets (β1 and β2) between TM10 and TM11 in our IF and IAI structures, switch their fold to an α-helix and loop, respectively, in the occluded conformation (Fig. 3b). In addition, TM10 extends to residue 390, compared to residue 368 in the IAI structures. The Coulombic electrostatic surfaces of the intracellular domain reveal that TM10 and helix I are positively charged, whereas the rest of the intracellular domain is more negatively charged (Fig. 3c).

Examination of the intracellular domain in the occluded conformation uncovered a phosphorylated Thr residue at position 410, encircled above by a pocket of positively charged Lys (Lys31, Lys328, Lys372, Lys461) and Arg (Arg327) residues at the cytosolic face of the transmembrane domain. The phosphate group of phosphorylated Thr410 interacts with this positive pocket (Fig. 4a, b), which is also well-defined in the IAI structure, even though the intracellular domain is not resolved in this conformation. Further comparison with the IAI conformation revealed an upward shift of Lys372 by approximately 5.4 Å due to a fold switch from a β-sheet to an α-helix. This repositioning allows the side chain of Lys372 to face Thr410 in the occluded conformation (Fig. 4c). Moreover, other positively charged residues also undergo a flip to orient themselves towards Thr410 in the occluded state, in contrast to their alignment in the IAI conformation (Fig. 4c).

To investigate the role of Thr410 phosphorylation, we mutated this residue to Val, whose side chain cannot be phosphorylated. AtBor1_active carrying this Thr410V mutation failed to rescue yeast growth on 80 mM boric acid (Fig. 4e and Supplementary Fig. 5b). Because Thr410 is highly conserved in plant BOR transporters (Fig. 4d), we propose that its phosphorylation is essential for borate transport, perhaps by contributing to the conformational change from IAI to occluded and/or stabilizing the core domain in the occluded conformation. However, the report of Thr410 phosphorylation requires further investigation before its role in borate transport can be fully established.

### AtBor1 operates with an elevator mechanism

Comparing our structures with published human SLC4 transporter structures in an outward-facing (OF) conformation (AE1 PDB: 7UZ3) indicates that AtBor1 transports substrate via the elevator mechanism like previously proposed (Fig. 5a and Supplementary Movies 2, 3)[28]. In the IF conformation, substrates in cells have access to the binding site

in the core domain, which then pivots vertically along the gate domain by approximately 4.2 Å to occluded conformation (Fig. 5 a, b). The core domain subsequently shifts further along the gate domain by approximately 3.3 Å to reach to the OF conformation, allowing substrates to exit the transporter to the extracellular milieu (Fig. 5 a, b). Additionally, the core domain undergoes a rotational movement of approximately 9 to 11 degrees (Fig. 5c). Throughout this elevator transport process, the gate domain remains rigid during elevator transport (Fig. 5a, b, c and Supplementary Fig. 11b).

Additionally, we also measured the vertical displacement of the well-characterized elevator transporter AE2, another SLC4 family member with known inward and outward conformations[34]. The vertical displacement for AE2 is approximately 7.9 Å. These findings indicate that the vertical movement of AtBor1 aligns with the range observed in other elevator-type transporters, thus supporting the consistency and relevance of our proposed model.

We identified a small cluster of four hydrophobic residues in the gate domain – Phe159, Ile163, and Phe167 on TM5 and Met499 on TM12 – which serve as a gate to cooperate with the substrate binding pocket in the core domain. In the IF conformation, the substrate binding pocket is located below these FIFM gating residues, facing the intracellular side to allow access to intracellular substrates (Fig. 5d and Supplementary Movie 4). When the core domain moves upwards to the occluded conformation, the substrate binding pocket directly faces the FIFM gating residues, which occlude the substrate binding pocket together with an additional hydrophobic residue on TM5, Leu166 (Fig. 5e and Supplementary Movie 4). To reach the OF conformation, the core domain moves further upwards to bring the substrate binding pocket above the FIFM gating residues, allowing substrates to exit the transporter (Fig. 5f and Supplementary Movie 4). These four hydrophobic residues are highly conserved in other SLC4 transporters thus we define this cluster as the FIFM gate (Supplementary Fig. 6).

## Discussion

We have described the structure of AtBor1 in an IAI, IF, and occluded conformation, as well as a hybrid IF/occluded conformation. Our IAI structure illustrates the molecular basis of an autoinhibitory mechanism in clade I BOR transporters. Replacement of the Ct hairpin in AtBor1 (clade I) with the corresponding sequence in AtBor4 (clade II) confirmed the inhibitory function of this structure. Because the physiological function of clade I BOR transporters is to transport borate from roots to shoots when boric acid is deficient in soil, this autoinhibitory mechanism is important for avoiding toxic accumulation of boron in plant tissues.

We propose that a similar mechanism may be responsible for functional regulation in other SLC4 transporters, including the human NDCBE-B/D transporter, whose Ct domain has an inhibitory effect on its activity[20], as well as the human AE2 transporter, whose recent cryo-EM structure revealed insertion of the Ct loop into the inner vestibule to occupy the anion-binding pocket[34]. Autoinhibitory activity has also been reported in the human NBCe1-b/c transporter and is associated with its unique N-terminus[21–27]. NBCe1-b/c can be fully activated by either removing its N-terminal autoinhibitory domain or adding the protein IRBIT (inositol 1,4,5-trisphosphate (IP3) receptor-binding protein released with inositol 1, 4, 5-trisphosphate)[21–27]. It is unclear how Bor1 autoinhibition is relieved, but we speculate that there may be a partner protein in plants that plays a similar role to IRBIT by releasing the autoinhibitory domain (Supplementary Fig. 11a). Previous studies in low boron conditions have reported that the clathrin adapter protein, adapter protein 2 complex (AP-2), recognizes residues 637-704 in the Ct of Bor1 to localize Bor1 with the correct polarity in the plasma membrane of root cells[45]. Because the start of this recognition site for AP-2 differs to that for the autoinhibitory domain by only one amino acid, we conjecture that AP-2 may associate with the Ct of Bor1 in low boron

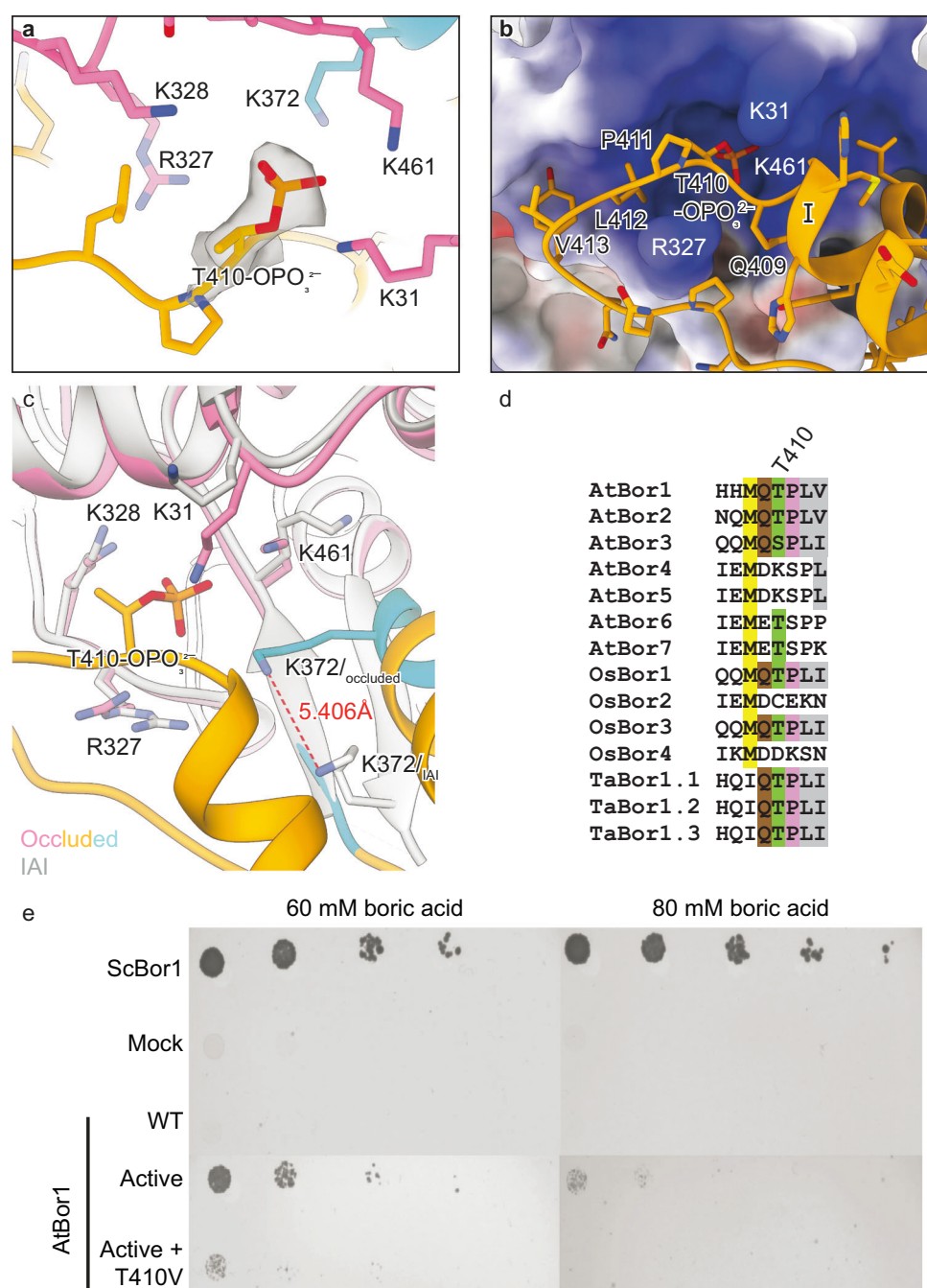

**Fig. 4 | Phosphorylated T410 plays a pivotal role in borate transport. a** Close-up view of phosphorylated T410 and the surrounding basic amino acid residues R327, K328, K372, K461, and K31 in the occluded conformation of AtBor1active. Cryo-EM density for phosphorylated T410 shown as gray surface. Residues from the core and intracellular domains are colored in pink and gold, respectively. The residue K372 from the intracellular domain, which forms the short β1 in the IAI conformation and switches to an α-helix in the occluded conformation, is colored in cyan. **b** Positioning of phosphorylated T410 in the intracellular domain (gold ribbons) within the positively charged pocket of the transmembrane domains (Coulombic electrostatic surfaces). Color scale as in (**a**). **c** Superimposition of the structures of inward-autoinhibited (gray) and occluded (colored) conformations. Core domains are superimposed. **d** Sequence alignment of the segment containing T410 in plant Bor transporters (At, *Arabidopsis thaliana*; Os, *Oryza sativa* (rice); Ta, *Triticum aestivum* (wheat)). **e** Yeast complementation assay shows that T410V mutation eliminates the activity of AtBor1active.

conditions to both maintain its polar localization and release the Ct autoinhibitory domain for borate transport. In addition, a recent study discovered that low boron levels can trigger the phosphorylation of Bor1 at its C-terminus, enhancing boron transport from roots to xylem[8]. Phosphorylation of the C-terminus may provide an alternative method for releasing the Ct autoinhibitory domain (Supplementary Fig. 11a). However, these proposed mechanisms require further investigation.

Several structures of SLC4 transporters have been solved in both OF and IF conformations[28–35,40,41]. However, a structure in an occluded conformation has not previously been reported for this family. We successfully captured the structures of AtBor1active in IF, occluded, and hybrid IF/occluded conformations. When considered with the published OF AE1 structure[31], our AtBor1 structures illustrate the molecular details of elevator transport in the SLC4 transporter family by providing the stepwise transitions in individual protomers that enable alternating access of the transporter (Supplementary Fig. 11b). Earlier functional studies on NBCe1-a concluded that each protomer in an SLC4 dimer has independent transport activity[44],

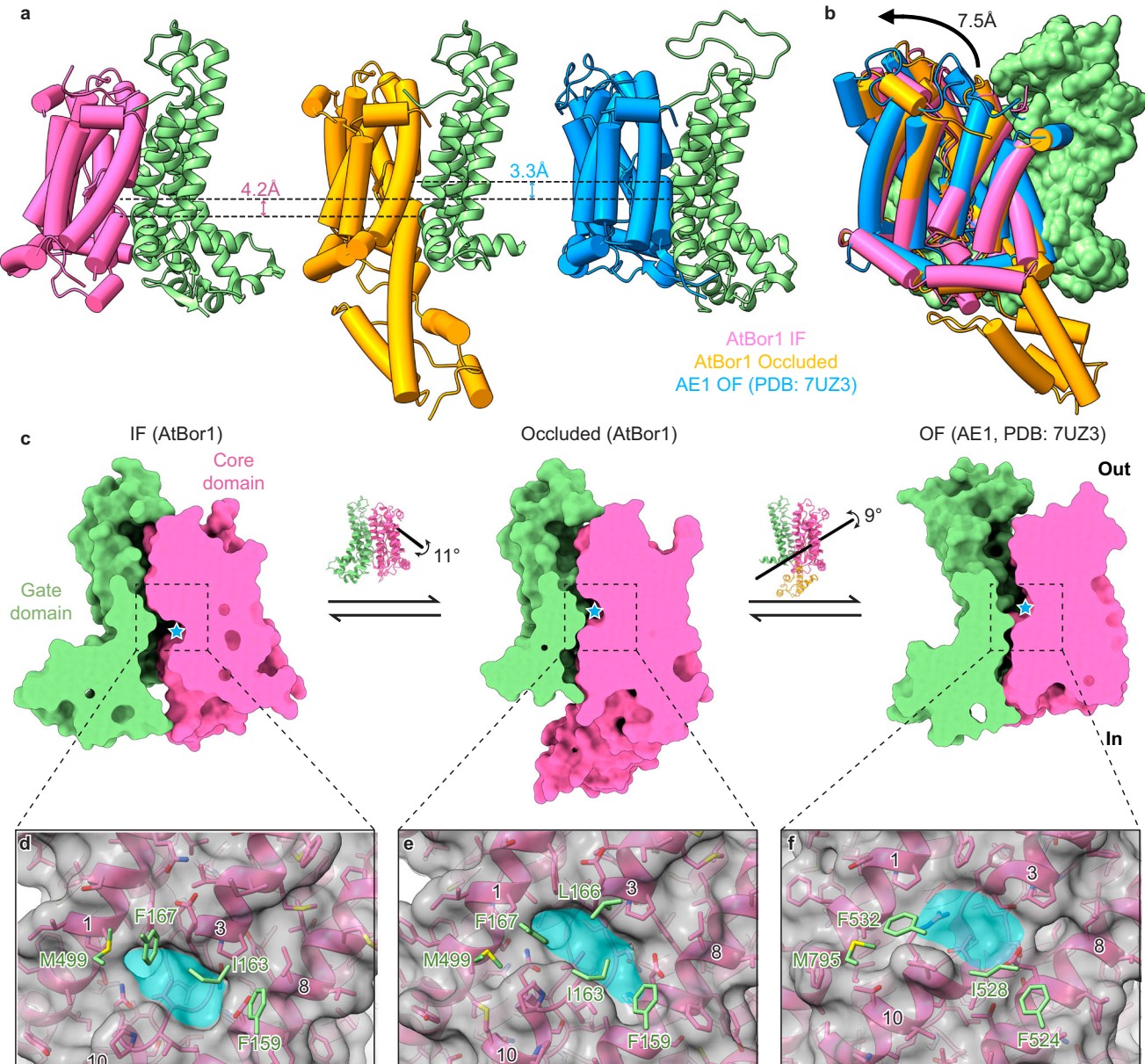

**Fig. 5 | Comparison of AtBor1 and other SLC4 structures reveals a conserved transport mechanism. a, b** Comparison of the core domain in inward-facing (AtBor1), occluded (AtBor1), and outward-facing (human AE1) conformations, aligned by the gate domain. The gate domain is colored in green, and the core domain is colored in hot pink, orange, or blue for IF (AtBor1), occluded (AtBor1), or OF (human AE1), respectively. **c** Side views of IF (AtBor1), occluded (AtBor1), and OF (human AE1) structures cut through the substrate coordination site (cyan star). Rotation of the core domain and rotation axes are shown on the AtBor1 ribbon representation insets. Note the different rotation axes for the conformational changes. **d**–**f** Close-up views of the substrate coordination site and the four conserved gating residues (F159, I163, F167, and M499 in AtBor1; F528, I528, F532, and M795 in human AE1) in the gate domain of inward-facing AtBor1 (**d**), occluded AtBor1 (**e**), and outward-facing AE1 (**f**).The core domain is shown as pink model and gray surface. The substrate coordination site is colored in cyan. Gating residues from the gate domain are colored in green.

consistent with our findings. Indeed, a recent cryo-EM structure of human AE2 showed a hybrid dimer with one protomer in an OF conformation and one in an IF intermediate conformation[34]. These studies collectively suggest that each monomer in an SLC4 family transporter can move and function independently (Supplementary Fig. 11b). Such stepwise transition of individual subunits has also been observed in another elevator transporter–the SLC1 glutamate transporter[46,47].

Importantly, our AtBor1 occluded structure resolved the large intracellular loop between TM10 and TM11. This region has been shown to be involved in the correct polar localization of Bor1 in the plasma membrane as well as endocytic degradation under high boron

conditions. Tyrosine residues on this loop have previously been proposed to serve as a tyrosine-based signal or a tyrosine phosphorylation site[15,48]. We did not observe phosphorylation of tyrosine residues but instead found phosphorylation of Thr410. Phosphorylated Thr410 interacts with a positive charge pocket at the cytosolic surface of the transmembrane domain, which contributes to the conformational change from IAI to occluded and/or stabilizing the core domain in the occluded conformation. We also showed its importance for the transport of borate using yeast complementation assays. The role of Thr410 in borate transport deserves further investigation given the conservation of this residue in AtBor1, 2, 6 and 7, OsBor1 and 3, and TaBor1.1-1.3 (Fig. 4d).

The experimental data we have presented provide a structural understanding of AtBor1 in inward-autoinhibited, inward-facing and occluded conformations, emphasizing the pivotal role of both the Ct and intracellular domain in regulating AtBor1 activity. Our results provide promise for improving crop yields in both boron-deficiency and boron-excess soils and enhancing our understanding of the regulation of transport mechanisms in the SLC4 transporter family.

## Methods

### Protein expression and purification

Nucleotides coding for the *Arabidopsis thaliana* borate transporter AtBor1 (UniProt ID: Q8VYR7) with a C-terminal deca-histidine tag was synthesized and code optimized for the *Saccharomyces cerevisiae* expression (Synbio Technologies). The expression fragment was cloned into a p423-GAL1 vector (ATCC® 87327™). *S. cerevisiae* yeast strain-INV*SC*1 (ThermoFisher) was used for the overexpression, and the transformation was conducted by the yeast transformation kit (Sigma). The protein overexpression and purification were followed by the published protocol until the step of the Ni-NTA purification[49]. Briefly, transformed yeast were cultured at 30 °C in CSM-His medium to an OD600 of approximately 10. Protein expression was induced by adding 8% (wt/vol) galactose dissolved in 4× yeast extract-peptone medium to reach a final concentration of 2% galactose. After 16 hours of shaking at 30 °C, cells were harvested by centrifugation at $4000 \times g$ for 15 minutes. Yeast pellets were resuspended in a buffer containing 50 mM Tris (pH 7.0), 1 mM EDTA, and 1 mM phenylmethylsulfonyl fluoride (PMSF). Cells were lysed using a high-pressure homogenizer, and the lysate was diluted with an equal volume of lysis buffer to achieve a final composition of 50 mM Tris (pH 7.0), 700 mM NaCl, 10% glycerol, 1 mM EDTA, and 1 mM PMSF. The lysate was centrifuged at $20,000 \times g$ for 10 minutes, followed by ultracentrifugation at $185,000 \times g$ for 1 hour to collect membrane fractions. The membranes were solubilized in a buffer containing 50 mM Tris (pH 7.0), 500 mM NaCl, 10% glycerol, and 1.5% (wt/vol) n-dodecyl-β-D-maltoside (DDM) for 1 hour at 4 °C. Unsolubilized material was removed by ultracentrifugation at $142,000 \times g$ for 30 minutes, and the solubilized sample was applied to pre-equilibrated Ni-NTA resin (Thermo Fisher). The resin was washed with 80 mM imidazole, and the protein was eluted with 300 mM imidazole. The elution from the Ni-NTA column was concentrated by a 50-kDa cutoff Amicon concentrator and further purified by the Superdex 200 Increase 10/300 GL gel filtration column equilibrated in 20 mM Tris pH 7.0, 100 mM Na₂SO₄, and 0.03% n-dodecyl- β-D-maltoside (DDM). The expression plasmid of the AtBor1 active mutant (R637E, E641R, R643E)−AtBor1$_{active}$ was generated by Takara Bio Site-Direction Mutagenesis Kit. The expression and purification details are same as the wild-type AtBor1. Protein was finally purified in the buffer containing 20 mM MES-NaOH pH6.5, 100 mM Na₂SO₄, 0.01% LMNG and 2 mM TCEP-HCl (final pH 6.5 adjusted by 1 M NaOH).

### Reconstitution of AtBor1 into lipid nanodiscs

The expression plasmid pMSP1E3 (a gift from Stephen Sligar (Addgene plasmid # 20064; http://n2t.net/addgene:20064; RRID:Addgene_20064)[50] was transformed into *E. coli*, BL21 (DE3) strain. The overexpressed protein was purified via Ni-NTA affinity chromatography as described[51] and then desalted in buffer containing 20 mM Tris-Cl, 0.1 M NaCl, 0.5 mM EDTA, pH7.4. 100 μL of 25 mg/mL 1-palmitoyl-2-oleoyl-glycero-3-phosphocholine lipids (16:0–18:1 PC, Avanti Polar Lipids, Inc.) was dried and resuspended in 200 μL of 20 mM sodium cholate. Purified AtBor1 in DDM, MSP1E3, and cholate solubilized PC were mixed in a molar ratio of 2 AtBor1: 2.5 MSP1E3: 183 PC. The protein mixture was incubated on a rotator at 4 °C for 1.5 h, then 0.03 g Bio-Beads SM2 (BioRad) was added per milliliter of incorporation sample. Sample with Bio-Beads was incubated on a rotator at 4 °C overnight. In the next day, Bio-Beads were removed from the sample

with low-speed centrifugation and the sample was further purified by the Superdex 200 Increase 10/300 GL gel filtration column in buffer containing 20 mM MES-NaOH pH 6.5, 100 mM Na₂SO₄, and 2 mM TCEP-HCl (final pH 6.5 adjusted by 1 M NaOH).

### Cryo-EM data acquisition

Freshly purified Bor1 protein in lipid nanodiscs was concentrated to ~3.5 mg/mL, then 3 μL of protein was placed on the glow-discharged holey carbon grids (Quantifoil Cu R1.2/1.3), which were blotted for 3 s and flash-frozen in liquid ethane cooled by liquid nitrogen with a Vitrobot Mark IV (Thermo Fisher Scientific Inc.). Grids preparation for AtBor1$_{active}$ in LMNG was same as the above except the blotting time for AtBor1$_{active}$ in LMNG was 5 s. The grids were subsequently transferred to a Titan Krios microscope operating at 300 kV equipped with GIF Quantum K2 system. CryoEM images were automatically collected via the program Leginon[52] at a nominal magnification of 165,000× (calibrated pixel size: 0.83 Å). Each exposure was dose-fractionated into 50 frames (8 frames/s) in the counting mode with a total dose of 55 e⁻/Å². Detailed data collection parameters are listed in Supplementary Table 1, 2.

### Cryo-EM data processing

The beam-induced motion within each dose-fractionated cryoEM micrograph was corrected using MotionCor2[53]. The defocus and astigmatism values of the micrographs were determined using CTFFIND4[54]. Particles were picked using Topaz[55]. For all data sets, the initial model was generated using the ab-initio reconstruction in cryoSPARC2[56]. The single particle data analysis was performed using RELION3[57] following customized workflows (Supplementary Fig. 2 and Supplementary Fig. 8). It is worth noting that different strategies of data processing were initially tested, which helped understand that nature of the data sets and design the final data processing strategies.

A total of 11,723 cryo-EM micrographs of wild-type AtBor1 were collected, and 7177 micrographs were selected for data processing by the results from CTFFIND4. 2,485,991 particles (2x binned, 144 × 144 pixels, 1.66 Å/pixel) of AtBor1 dimers picked by Topaz were directly sent to 3D classification with C2 symmetry using an ab-initio model generated by cryoSPARC2. The 1,121,358 particles from the best 3D class out of 4 classes were re-extracted in 288 × 288 pixels without binning (0.83 Å/pixel) followed by steps of 3D auto-refinement, particle polishing, 2D classification, and CTF refinement to achieve a reconstruction at 2.3 Å resolution with C2 symmetry implied using 1,012,376 particles (Supplementary Fig. 2). To further improve the resolution that was limited by the imperfect symmetry of the AtBor1 dimer, additional 3D classification and 3D auto-refinement focused on a single protomer of the dimer were performed using the following steps. The particles from the 3D-autorefinement were symmetry-expanded with the C2 symmetry using *relion_particle_symmetry_expand* followed by moving the particle centers to the center of the protomer using *relion_star_handler*. The expanded particles were subjected to further 3D classification, 3D auto-refinement, particle polishing, and CTF refinement to achieve a final reconstruction at 2.2 Å resolution using 853,837 asymmetric units. Both structures of the dimer and the protomer were found to be in the identical inward-facing conformation. To avoid the mismatch of size and shape between the reference of the small protomer and the particle images of the large dimer during 3D classification and 3D auto-refinement, a special masking method with composite masks were used in a modified version of RELION3 as follows[58]. In additional to the regular reference mask (magenta mask in Supplementary Fig. 2) that covered only a protomer, a second mask (gray mask in Supplementary Fig. 2) that covered the other protomer and the lipid nanodiscs were used and its covered regions were low-pass filtered to 20 Å during 3D classification and 3D auto-refinement. When this method with composite masks is used, the low-resolution information of the whole particle

helps constrain the alignment globally and signals of the protomer contribute more to 3D classification and 3D auto-refinement at high-resolution and finer alignment.

A total of 11,586 cryo-EM micrographs of AtBor1$_{active}$ were collected and 8816 micrographs were selected for data processing by the results from CTFFIND4. 4,547,488 particles (2x binned, 144 × 144 pixels, 1.66 Å/pixel) of dimers picked by Topaz were screened by several runs of 2D classification. 1,200,507 particles were selected for ab-initio reconstruction using cryoSPARC2 and for 3D auto-refinement with C2 symmetry that yielded a reconstruction at 3.4 Å resolution which reached the Nyquist limit (pixel size 1.66 Å). The selected particles were re-extracted in 288 × 288 pixels without binning (0.83 Å/pixel) and sent to 2D classification and 3D auto-refinement with C2 symmetry that yielded 3.0 Å resolution using 760,712 particles (Supplementary Fig. 8). To perform 3D classification and 3D auto-refinement of a single protomer in the AtBor1 mutant dimer, a procedure of symmetry expansion and using composites masks, similar to that for the above data processing of wild type AtBor1, was conducted. 1,521,424 asymmetric units after symmetry expansion were sorted into 4 classes by 3D classification with composite masks. The best class with 485,896 asymmetric units were subjected to steps of 3D auto-refinement, particle polishing, 2D classification, and CTF refinement that finally yielded a reconstruction of the protomer at 2.6 Å resolution using 479,691 asymmetric units (Supplementary Fig. 8). This structure of the protomer was found to be in the occluded state in contrast to the inward-autoinhibition conformation of the wild-type AtBor1 structure solved above. The asymmetric units from the other three 3D classes were combined and sent to another run of 3D classification that revealed a class of 433,512 asymmetric units in the inward-facing conformation. The asymmetric units in the inward-facing conformation and in the occluded conformation from these two runs of 3D classification were combined and the particles of dimers with both protomers selected were assigned in three groups: inward-facing dimers (80,443 particles), occluded dimers (110,420 particles), and inward-facing/occluded hybrid dimers (132,093 particles). Those particles of dimers with only one protomer selected were not further used. These three groups of particles were separately processed as dimers using 3D auto-refinement, particle polishing, and CTF refinement and yielded a reconstruction at 3.0 Å (inward-facing dimer), 2.8 Å (occluded dimer), and 3.0 Å (hybrid inward-facing/occluded dimer) resolution, respectively (Supplementary Fig. 8). C2 symmetry was implied for the inward-facing dimer and occluded dimer. No symmetry was implied for the hybrid inward-facing/occluded dimer because it is an asymmetric structure.

The modified source code of RELION with implementation of composite masks[58] is available at GitHub https://github.com/jiangjiansen/relion_composite_masks.

### Model building

Models of AtBor1 in different conformations were manually built from scratch using Coot[59]. Models were built based on the monomer maps due to the better side chain density and connectivity. Two monomer models were fitted into the dimer map and merged. AtBor1 dimer models were further checked and refined in the dimer maps. PHENIX[60] was used to do the real space refinement, and models were validated by comprehensive validation in PHENIX. Structural figures were generated using PyMOL and UCSF Chimera[61].

### Yeast Complementation assay

The expression plasmids of the AtBor1 with either truncated or mutant C-terminus was generated by Q5® Site-Directed Mutagenesis Kit (NEB) and Takara Bio Site-Direction Mutagenesis Kit. The primers used for the mutagenesis are listed in Supplementary Table 3. Nucleotides encoding for *S. cerevisiae* boron transporter ScBor1 (UniProt ID: P53838) was synthesized and cloned into the p423-GAL1 vector

(Synbio Technologies). *S. cerevisiae* Y01169 (*ScBOR1Δ*) strain with the following genotype: MATa; ura3Δ0; leu2Δ0; his3Δ1; met15Δ0; YNL275w:kanMX4 was purchased from Euroscarf and used in the complementation assay. Cells were transformed with the expression plasmids of wild type AtBor1, C terminus truncated or mutant AtBor1, ScBor1 or the empty p423-GAL1 vector using the yeast transformation kit (Sigma). Yeasts carrying different DNA plasmids were cultured in CSM-His media containing 2% galactose at 30 °C, shaking for ~20 hours. The yeast culture was then diluted with the CSM-His media to make OD$_{600}$ to 1.0 as a start point. 5 µL of a dilution series (1, 1/5, 1/25, 1/125 and 1/625) were spotted on CSM-His plates supplemented with 2% galactose containing 0, 40 mM, 60 mM or 80 mM boric acid. The plates were incubated at 30 °C for 6 days and pictures were then taken.

### Reporting summary

Further information on research design is available in the Nature Portfolio Reporting Summary linked to this article.

## Data availability

The data that support this study are available from the corresponding authors upon request. The cryo-EM maps have been deposited in the Electron Microscopy Data Bank (EMDB) under accession codes EMD-41185 (AtBor1 protomer), EMD-41186 (AtBor1 dimer), EMD-41188 (AtBor1$_{active}$ occluded protomer), EMD-41190 (AtBor1$_{active}$ occluded dimer), EMD-41191 (AtBor1$_{active}$ IF dimer) and EMD-41192 (AtBor1$_{active}$ occluded/IF dimer). Atomic model coordinates have been deposited in the Protein Data Bank (PDB) under accession codes 8TEG (AtBor1 protomer), 8TEH (AtBor1 dimer), 8TEJ (AtBor1$_{active}$ occluded protomer), 8TEL (AtBor1$_{active}$ occluded dimer), 8TEM (AtBor1$_{active}$ IF dimer), 8TEN (AtBor1$_{active}$ occluded/IF dimer). The previously published structural model of AE1 used in this research is available from the PDB under accession code 7UZ3. The source data underlying the protein purification chromatographs in Supplementary Figs. 1a and 7a, the uncropped SDS-PAGE gels in Supplementary Figs. 1b and 7b, and the uncropped images of the agar plates from the yeast complementation assay (Figs. 2g, 4e, and Supplementary Fig. 5) are provided in the Source Data file. Source data are provided with this paper.

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

## Acknowledgements
This work was supported by the Intramural Research Program at National Heart, Lung, and Blood Institute (NHLBI), the National Institutes of Health (NIH). This work utilized the NIH Multi-Institute Cryo-EM Facility (MICEF) and the NIH Biowulf high-performance computing systems (http://hpc.nih.gov). We thank Yi He for providing access to the high-pressure homogenizer in the Fermentation Facility. We thank Huaibin Wang and Haifeng He for technical support on the electron microscopes. Yan Jiang is an affiliate member of the Australian Research Council Industrial Transformation Training Center for Cryo-electron Microscopy of Membrane Proteins (IC200100052).

## Author contributions
Y.J. and J.J. conceptualized and planned the study. Y.J. performed the experiments. Y.J. and J.J. analyzed the data and wrote the manuscript.

## Funding

## Competing interests
The authors declare no competing interests.
