## [Transparent Peer Review file · Nature Communications]

The Bor1 elevator transport cycle is subject to autoinhibition and activation

Corresponding Author: Dr Yan Jiang

Version 0:

Reviewer comments:

Reviewer #1

(Remarks to the Author)

Overall comments:

Arabidopsis thaliana Bor1 is the first discovered borate transporter, and possesses close homology with human SLC4 transporters whose malfunctions are the molecular causes of many diseases. AtBor1 had previously had its structure determined by low-resolution X-ray crystallography, which led to a proposal of an elevator transport mechanism for it and transporters in the SLC4 family. In this work, the authors present high-resolution cryoEM structures of AtBor1 in multiple states: inward-autoinhibited, inward-facing, occluded, and a hybrid inward/occluded structure. The structural data are of high quality, with quality maps shown and strong stats on their built models. These data have the potential to provide three main new valuable new insights to the field: (1) the role of a C-terminal region in autoinhibition, (2) the role of phosphorylation of Thr410 in activating transport, and (3) depicting multiple conformational states of the AtBor1 transport cycle thereby supporting an elevator mechanism. However, I have questions about the analysis and presentation of the data that I think the authors need to address before the paper can be published. I have outlined my comments as either major or minor and present them below:

Major comments:

Line 97: The authors cite 14 papers (citations 27 through 40) with which their AtBor1 structures share structural homology, and in all 14 of those papers the structures are described using "Gate" and "Core" domain nomenclature to describe the protein's two major domains. Yet in this paper, the authors label the Gate as the "scaffold" and Core as the "carrier." Given the critical mass of papers already adopting the Gate and Core terminology, it is confusing for the authors to use a new nomenclature, especially without any acknowledgement or definition of it. The authors should use the original nomenclature, or else the paper is at risk of being isolated from and harder to understand than the prior papers.

Figure 1: It is not stated in the figure, the figure legend, or in the accompanying text what state the authors think this structure in Fig. 1 represents: open outward, open inward, occluded, etc. The authors need to specify and make that identification explicit. Presumably it is inward and inhibited, but that is not discussed until much later in line 182. Make that clear earlier, including in the figure and its caption.

Figure 2: The findings in the Ct region are interesting and provocative, but I am curious why the authors did not test an R637T mutant? They show that in the conserved TRSRGE motif, the only difference between Clade 1 and Clade 2 BOR members is the shift of RT. Might that mutation alone be sufficient to enable growth when introduced into AtBor1? Why was only the RA shown? And similarly, why in line 174 was a full 10-amino acid stretch, encompassing more than just the TRSRGE motif, chosen for the AtBor4 sequence to be placed into AtBor1? Is it because that 10-amino acid region contains other amino acids that are important besides TRSRGE? If so, which ones? There is an opportunity here for the authors to offer clarity and be more specific in not only which amino acids constitute the autoinhibition, but also what they think are the critical differences in sequences here between the Clade 1 and Clade 2 transporters, something which has remained mysterious in the field. Relatedly, in discussing these points the authors should cite paper PMID: 35903773 which looks at the ability of all 7 AtBOR transporters to complement in the yeast genetic assay. All Clade 1 members failed to grow while all Clade 2 members succeeded with the exception of AtBor6. Do your data explain these observations, possibly including whether there is an important difference in the sequence of AtBor6?

Line 245: The authors indicate degree of rotation, but it should also be stated how much vertical transposition through the membrane the Core(carrier) domain would undergo in these transitions, and compare that number with the measurement of vertical transposition for other known elevator transporters. Fig 5c makes the vertical transposition looks small, but it should be quantified and discussed and contextualized with other elevator transporters, which I think will show that it is on the smaller side of vertical movements but still consistent with those shown for some other elevator transporters.

Fig 5: There are important issues to be addressed with the presentation of data in Figure 5. For Fig 5a: Why were the Core(carrier) and Gate(scaffold) domains superposed separately in panel a? Unsurprisingly, when doing so, they all would have very small RMSD's and show little variation, which is exactly what we see here. Figure 5a is never mentioned in the main text of the paper, presumably because it does not reveal any new or valuable insights. The papers previously cited have already well established how closely related all these proteins are. I am not convinced Fig 5a adds any value to the manuscript and should be removed unless the authors can offer a rationale and incorporate that rationale into the text. A stronger idea is that, given the mostly static nature of the Gate(scaffold) domains from prior work, simply superpose structures based *just* on the Gate(scaffold) domains while showing how the Core(carrier) domains move relative to such a superposition, which would highlight the molecular motions that AtBor1 and other transporters undergo. In Fig 5b the authors attempt to do exactly this, but the figure poorly represents what the authors are trying to show. The differences in either rotation or vertical transposition of the Core(carrier) domain are not apparent from looking at this figure. Rather than revealing key differences, the figure obscures them and makes Core/carrier positions all look similar, which undermines the points the authors are trying to make with respect to the elevator mechanism. (Additionally, the blue/green/light blue color scheme is not a good choice for revealing differences between structures, either). This figure must be re-imagined if it is to show these differences effectively. I do not doubt that the differences exist, and support an elevator model, but that is not what this figure shows. Video 3, on the other hand, does an excellent job in showing these differences between conformational states. The text and figures here need to match the quality of that video, where structures corresponding to different conformational states have their differences readily identifiable.

Minor comments:

The authors refer to Bor1 as an SLC4 transporter. SLC4 transporters have distinct functional, evolutionary, and nomenclature differences from the borate transporters. It is therefore more accurate to say that Bor1 shares homology with SLC4 transporters, rather than being bona fide members of the SLC4 family. My view is that this language should be reflected throughout the manuscript.

Line 36: The first sentence of the intro ought to cite the original Warrington 1923 paper:
<https://doi.org/10.1093/oxfordjournals.aob.a089871>

Line 72: I don't think the phrase "long-sought-after" is merited here.

Figure 2f: Indicate numbering for AtBor1 in this alignment.

Figure 4: The data on the Thr410 are interesting, but was the only T410V mutant made in the AtBor1 (active) background, which has an additional three mutations? Did the authors ever test the effect of just the T410V mutation in the AtBor1 (delta627) construct?

Closely proofread the manuscript, as there are typographical errors.

Reviewer #2

(Remarks to the Author)

This manuscript reports cryo-EM structures of atBor1 in both inward-facing and occluded states. The high-resolution structures allow for the visualization of previously unobserved structural details. The inward-facing state of atBor1 reveals an autoinhibitory domain at the C-terminal, which blocks the substrate permeation pathway. The occluded state of atBor1 reveals the intracellular domain, which is disordered in the inward-facing state, and uncovers a phosphorylated Thr410, contributing to the conformational change from the inward-facing to the occluded state. Functional analysis further confirms the inhibitory effect of the C-terminal domain and the activation effect of Thr410 phosphorylation. This study is well-performed, and the manuscript is well-written.

Specific comments:

1. Line 176-178: The results described on lines 173-175 only show that atBor4 is more active than atBor1, which rescued yeast growth in 80 mM boric acid. There is no evidence that the activity of atBor1 is inhibited under high boron conditions.
2. Fig2c-e: The nitrogen in AID is hard to see. Perhaps using a different color for AID would help.
3. Line 196: AtBor1 transports substrates via an elevator mechanism. The scaffold domain remains stable during the conformational change of other elevator mode SLC4 family transporters. Why are TM13 and TM14 in the scaffold domain of atBor1 disordered in the occluded state?

Version 1:

Reviewer comments:

Reviewer #1

(Remarks to the Author)

The authors have responded to the comments in a satisfactory way, and I believe all the major questions have been resolved, especially a much stronger Figure 5. I think this paper will be highly significant to its field and those of related transporters.

Reviewer #2

(Remarks to the Author)

The manuscript is improved, and my major concerns are addressed.

We sincerely thank the reviewers for recognizing the quality and significance of our work and for their valuable constructive comments and suggestions which have not only enhanced our manuscript but also reinforced the strength of our scientific findings. In the revised manuscript, we have addressed all comments from the reviewers. Below, we have provided point-to-point responses that address each of the reviewers' comments.

Reviewer #1 (Remarks to the Author):

Overall comments:

Arabidopsis thaliana Bor1 is the first discovered borate transporter, and possesses close homology with human SLC4 transporters whose malfunctions are the molecular causes of many diseases. AtBor1 had previously had its structure determined by low-resolution X-ray crystallography, which led to a proposal of an elevator transport mechanism for it and transporters in the SLC4 family. In this work, the authors present high-resolution cryoEM structures of AtBor1 in multiple states: inward-autoinhibited, inward-facing, occluded, and a hybrid inward/occluded structure. The structural data are of high quality, with quality maps shown and strong stats on their built models. These data have the potential to provide three main new valuable new insights to the field: (1) the role of a C-terminal region in autoinhibition, (2) the role of phosphorylation of Thr410 in activating transport, and (3) depicting multiple conformational states of the AtBor1 transport cycle thereby supporting an elevator mechanism. However, I have questions about the analysis and presentation of the data that I think the authors need to address before the paper can be published. I have outlined my comments as either major or minor and present them below:

Major comments:

Line 97: The authors cite 14 papers (citations 27 through 40) with which their AtBor1 structures share structural homology, and in all 14 of those papers the structures are described using "Gate" and "Core" domain nomenclature to describe the protein's two major domains. Yet in this paper, the authors label the Gate as the "scaffold" and Core as the "carrier." Given the critical mass of papers already adopting the Gate and Core terminology, it is confusing for the authors to use a new nomenclature, especially without any acknowledgement or definition of it. The authors should use the original nomenclature, or else the paper is at risk of being isolated from and harder to understand than the prior papers.

We thank the reviewer for pointing out this issue. While we believe "scaffold" and "carrier/transport" better reflect the functional roles of the two domains in AtBor1 and other SLC4 transporters and they have been used in previous reports such as the Bor1p structure paper (Coudray et al., PMID: 15192975), we fully agree with the reviewer on the importance of adhering to established nomenclature. As such, we have updated the terms to "gate" and "core" throughout the manuscript.

Figure 1: It is not stated in the figure, the figure legend, or in the accompanying text what state the authors think this structure in Fig. 1 represents: open outward, open inward, occluded, etc. The authors need to specify and make that identification explicit. Presumably it is inward and inhibited, but that is not discussed until much later in line 182. Make that clear earlier, including in the figure and its caption.

We have clarified the conformation in Fig. 1, its caption, and the relevant main text, as suggested by the reviewer.

Figure 2: The findings in the Ct region are interesting and provocative, but I am curious why the authors did not test an R637T mutant? They show that in the conserved TRSRGE motif, the only difference between Clade 1 and Clade 2 BOR members is the shift of RT. Might that mutation alone be sufficient to enable growth when introduced into AtBor1? Why was only the RA shown?

The R637A mutant presented here was part of a broader alanine-scanning mutagenesis study to investigate the role of each residue within the TRSRGE motif. Replacing R637 with alanine (or glutamate in the R637E/E614R/R643E triple mutant) disrupts the salt bridge between R637 and E502, resulting in AtBor1 activation. While we lack data on the R637T mutation specifically, we anticipate it would also lead to AtBor1 activation by similarly disrupting this salt bridge.

It's important to note that this R to T substitution is not the sole difference in the autoinhibition domain between Clade 1 and Clade 2 BOR members. The residue preceding T636 (M/I/F in AtBor1-3, respectively) is absent in Clade 2 (AtBor4-7), which could also contribute to disrupting autoinhibition domain binding.

Our combined results from single point mutations and deletions demonstrate that the entire TRSRGE motif (along with other highly conserved residues in the autoinhibition domain) is essential for autoinhibition. Mutating any single residue within the TRSRGE motif (except for S638A) is sufficient to abolish this function, suggesting a coordinated mechanism involving multiple conserved residues rather than a single determinant.

And similarly, why in line 174 was a full 10-amino acid stretch, encompassing more than just the TRSRGE motif, chosen for the AtBor4 sequence to be placed into AtBor1? Is it because that 10-amino acid region contains other amino acids that are important besides TRSRGE? If so, which ones?

We appreciate the reviewer's question, allowing us to clarify the definition of the TRSRGE motif. It encompasses a 9-residue stretch (T636-H644; TRSRGEFRH in AtBor1) (Fig. 2f). We named this motif based on the highly conserved TRSRGE sequence within it. This definition is now explicitly stated in the revised manuscript (line 132). When replacing AtBor1's TRSRGE motif with that of AtBor4, we also included the preceding M635 residue, whose potential role in autoinhibition was discussed earlier. Thus, a 10-residue segment in AtBor1 was replaced with the corresponding region from AtBor4 in this specific mutant.

There is an opportunity here for the authors to offer clarity and be more specific in not only which amino acids constitute the autoinhibition, but also what they think are the critical differences in sequences here between the Clade 1 and Clade 2 transporters, something which has remained mysterious in the field. Relatedly, in discussing these points the authors should cite paper PMID: 35903773 which looks at the ability of all 7 AtBOR transporters to complement in the yeast genetic assay. All Clade 1 members failed to grow while all Clade 2 members succeeded with the exception of AtBor6. Do your data explain these observations, possibly including whether there is an important difference in the sequence of AtBor6?

The autoinhibition domain (AID) comprises H12 and the TRSRGE motif (Fig. 2f), which are both crucial for autoinhibition function. In the inward-autoinhibited AtBor1 structure, the

TRSRGE motif acts as a plug inserting into the substrate permeation pathway, and H2 interacts with the transporter periphery (Fig. 2b and e). Cryo-EM and mutagenesis data indicate that most AID residues are essential for autoinhibition, and mutating even a single conserved residue can disrupt this function.

Sequence alignment of the AID region from Arabidopsis, rice, and wheat BOR transporters reveals two distinct patterns at the N-terminus of the TRSRGE motif: “XTR” in Clade 1 and “-TT” in Clade 2 (X represents M/I/F/V; - denotes a missing residue). This 3-residue segment may be key in differentiating the two clades. However, without a broader analysis beyond the AID, it’s uncertain whether other regions also contribute to functional or regulatory differences between the clades.

We currently lack data to explain why AtBor6 failed to rescue yeast growth in the previous complementation assay (PMID: 35903773). Sequence alignment reveals a few unique residues in AtBor6 compared to other Clade 2 transporters, which could be responsible. Alternatively, AtBor6 activity may require additional regulatory mechanisms. We agree this is an intriguing question deserving further investigation. The suggested paper (PMID: 35903773) is now properly cited in the revised manuscript (lines 165-168).

Line 245: The authors indicate degree of rotation, but it should also be stated how much vertical transposition through the membrane the Core(carrier) domain would undergo in these transitions, and compare that number with the measurement of vertical transposition for other known elevator transporters. Fig 5c makes the vertical transposition looks small, but it should be quantified and discussed and contextualized with other elevator transporters, which I think will show that it is on the smaller side of vertical movements but still consistent with those shown for some other elevator transporters.

We agree with reviewer on the importance of quantifying the Core domain movement and comparing it to other SLC4 transporters. The vertical displacement is now reported in the revised manuscript (lines 246-249) and new Fig. 5a and b. Specifically, the Core domain shifts 4.2 Å vertically from the inward-facing to occluded conformation, and an additional 3.3 Å from the occluded to the published outward-facing AE1 structure (PDB: 7UZ3), totaling 7.5 Å. We also measured the vertical displacement in AE2, another well-characterized SLC4 elevator transporter with available inward and outward structures (PMID: 37002221), which is approximately 7.9 Å. Thus, the observed movement in Bor1 falls within the range seen in other SLC4 transporters with an elevator transport mechanism, further supporting our model’s validity.

Fig 5: There are important issues to be addressed with the presentation of data in Figure 5. For Fig 5a: Why were the Core(carrier) and Gate(scaffold) domains superposed separately in panel a? Unsurprisingly, when doing so, they all would have very small RMSD’s and show little variation, which is exactly what we see here. Figure 5a is never mentioned in the main text of the paper, presumably because it does not reveal any new or valuable insights. The papers previously cited have already well established how closely related all these proteins are. I am not convinced Fig 5a adds any value to the manuscript and should be removed unless the authors can offer a rationale and incorporate that rationale into the text. A stronger idea is that, given the mostly static nature of the Gate(scaffold) domains from prior work, simply superpose structures based *just* on the Gate(scaffold) domains while showing how the Core(carrier) domains move relative to such a superposition, which would highlight the molecular motions that AtBor1 and other transporters undergo. In Fig 5b the authors attempt to do exactly this,

but the figure poorly represents what the authors are trying to show. The differences in either rotation or vertical transposition of the Core(carrier) domain are not apparent from looking at this figure. Rather than revealing key differences, the figure obscures them and makes Core/carrier positions all look similar, which undermines the points the authors are trying to make with respect to the elevator mechanism. (Additionally, the blue/green/light blue color scheme is not a good choice for revealing differences between structures, either). This figure must be re-imagined if it is to show these differences effectively. I do not doubt that the differences exist, and support an elevator model, but that is not what this figure shows. Video 3, on the other hand, does an excellent job in showing these differences between conformational states. The text and figures here need to match the quality of that video, where structures corresponding to different conformational states have their differences readily identifiable.

We appreciate the reviewer's feedback on Fig. 5a and b. We have revised these panels accordingly, superimposing the Gate domains and highlighting the Core domain movements for clarity. We have also used more distinct colors to enhance visual differentiation between the structures. Additionally, we have included details in the text regarding the Core domain movement and its comparison to other SLC4 transporters.

Minor comments:

The authors refer to Bor1 as an SLC4 transporter. SLC4 transporters have distinct functional, evolutionary, and nomenclature differences from the borate transporters. It is therefore more accurate to say that Bor1 shares homology with SLC4 transporters, rather than being bona fide members of the SLC4 family. My view is that this language should be reflected throughout the manuscript.

We have followed the reviewer's recommendation and written in the Abstract and Introduction that Bor1 is homologous to the SLC4 transporters.

Line 36: The first sentence of the intro ought to cite the original Warington 1923 paper: <https://doi.org/10.1093/oxfordjournals.aob.a089871>

We thank the reviewer for highlighting this earlier work, which is now cited in the revised manuscript.

Line 72: I don't think the phrase "long-sought-after" is merited here.

We have deleted this word.

Figure 2f: Indicate numbering for AtBor1 in this alignment.

Done.

Figure 4: The data on the Thr410 are interesting, but was the only T410V mutant made in the AtBor1(active) background, which has an additional three mutations? Did the ever test the effect of just the T410V mutation in the AtBor1(delta627) construct?

It's correct that T410V mutant was generated in the context of active AtBor1 R637E/E641R/R643E triple mutant. We did not examine the T410V mutation in other active forms of AtBor1, such as delta627.

Closely proofread the manuscript, as there are typographical errors.

We and a colleague have carefully proofread the manuscript to eliminate some typos.

Reviewer #2 (Remarks to the Author):

This manuscript reports cryo-EM structures of atBor1 in both inward-facing and occluded states. The high-resolution structures allow for the visualization of previously unobserved structural details. The inward-facing state of atBor1 reveals an autoinhibitory domain at the C-terminal, which blocks the substrate permeation pathway. The occluded state of atBor1 reveals the intracellular domain, which is disordered in the inward-facing state, and uncovers a phosphorylated Thr410, contributing to the conformational change from the inward-facing to the occluded state. Functional analysis further confirms the inhibitory effect of the C-terminal domain and the activation effect of Thr410 phosphorylation. This study is well-performed, and the manuscript is well-written.

Specific comments:

1. Line 176-178: The results described on lines 173-175 only show that atBor4 is more active than atBor1, which rescued yeast growth in 80 mM boric acid. There is no evidence that the activity of atBor1 is inhibited under high boron conditions.

We acknowledge the reviewer's point about the conclusion at lines 176-178 being speculative. We have rephrased it in the revised manuscript (lines 178-180) to:

“These results establish a novel Ct autoinhibitory mechanism in clade I transporters that allows them to dynamically adjust transport activity in response to varying boron conditions, thus safeguarding shoots from both boron deficiency and toxicity.”

2. Fig2c-e: The nitrogen in AID is hard to see. Perhaps using a different color for AID would help.

The AID in Fig. 2c-e is now colored light blue for better visibility.

3. Line 196: AtBor1 transports substrates via an elevator mechanism. The scaffold domain remains stable during the conformational change of other elevator mode SLC4 family transporters. Why are TM13 and TM14 in the scaffold domain of atBor1 disordered in the occluded state?

It is unclear why TM13 and TM14 are invisible in the occluded state. If we place TM13 and TM14 to the same location as in the inward-facing state, the carrier (core) domain in the occluded state would clash with TM14, suggesting that the scaffold (gate) domain does not remain rigid and some flexibilities are required for TM14 and associated TM13 during the transport cycle. Instead of becoming completely disordered, these transmembrane helices likely undergo minor movements and become more dynamic in the occluded state, leading to their invisibility in the cryo-EM 3D reconstruction.